# OpenReview forum: "MMSU: A Massive Multi-task Spoken Language Understanding and Reasoning Benchmark"
_ICLR.cc/2026/Conference — ICLR 2026 Poster_

### Official Review · Reviewer_oqyG · 2025-10-23

**Soundness:** 3
**Presentation:** 4
**Contribution:** 3
**Rating:** 6
**Confidence:** 4

**Summary:**

This paper proposed a comprehensive benchmark for evaluating Speech LLMs on perception and reasoning tasks. They end up curating a total of 47 distinct tasks with 5000 audio-question-answer triplets. Their taxonomy and definition of tasks are expert-grounded, including both paralinguistics (emotion, speed, pitch) and linguistics (phonetics, prosody, rhetoric, syntactics, semantics). They also evaluate 22 Speech LLMs on the proposed benchmark and provide some insights and shortcomings of current SoTA models.

**Strengths:**

1. I deeply appreciate the efforts the authors have made for curating such a comprehensive audio benchmark with a reasonable taxonomy, and I believe this is beneficial to the future speech/audio community.
2. Compared to prior works on audio benchmarks, their evaluation tasks are more diverse and more organized. Also, it employs more real speech/audio for evaluation.
3. They evaluated a massive amount of Speech LLMs on their proposed benchmark.

**Weaknesses:**

1. I don’t fully understand why Speech LLMs are generally bad at perception tasks but good at complex reasoning tasks. Shouldn’t the model have to be able to perceive the acoustic cues and then do the reasoning? Or simply because Speech LLMs are good at “guessing” the answer? Or because the curated reasoning tasks are too easy?
2. Following the first point, why does Qwen2.5-Omni-7B outperform humans by 6% on Reasoning Linguistic (Semantics)? Is there anything wrong with the curated eval data for this task?
3. The selection of models for evaluation is somewhat less informative. I’m not sure why the author chose this selection of models. From Table 1, it seems like all models have similar size parameters (7~10B), and some of the models are quite old. I think readers would be more interested in recent models and also models that are diverse in different sizes. For instance, I think Voxtral Small 24B (https://arxiv.org/abs/2507.13264) or StepAudio 130B (https://arxiv.org/abs/2502.11946) might be a good option to include. Or other suitable open source models with larger size.

**Questions:**

1. It would be helpful if the author could refer to the section of the Appendix rather than just refer to the entire Appendix in the main manuscript.
2. For Table 1, I think it would be beneficial to know the text/speech tokens or instruction samples these models are trained on. I hypothesize that it is quite related to the performance, even though most models in the table have a similar number of parameters. It’s just a suggestion, I can fully understand that you might have some space constraints for the paper, and this is not the core information of the paper.

---

> ### Author Response · Authors · 2025-11-21
> **Author Response to Reviewer oqyG (1/2)**
>
> Dear Reviewer oqyG:
>
> Thank you for the valuable comments and encouraging feedback. We appreciate your recognition of our contributions. We address the specific comments below:
>
> **W1:  Why Speech LLMs are generally bad at perception tasks but good at complex reasoning tasks?**
>
> Thank you for the thoughtful question. Our results reflect a well-known pattern in current SpeechLLMs: perception and reasoning rely on different capabilities, and the former remains the bottleneck. Perception tasks in MMSU require precise recognition of fine-grained acoustic cues—prosody, stress, phonetic contrasts—which current SpeechLLMs often fail to understand. In contrast, many complex reasoning tasks in MMSU depend less on low-level acoustic discrimination and more on semantic inference, world knowledge, and textual reasoning, areas where modern LLMs are strong. This explains why SpeechLLMs can perform well on reasoning even when acoustic perception is imperfect—this is not “guessing,” but rather the model leveraging its strong language-reasoning priors to compensate for missing acoustic detail.
>
> Importantly, the reasoning tasks are not trivial. Human accuracy on several reasoning subtasks is far from perfect due to factors such as attention lapses, unfamiliar content, and subtle contextual cues, which reflect the complexity nature of these reasoning tasks. This further confirms that the strong performance of SpeechLLMs is driven by genuine reasoning capabilities rather than question simplicity.
>
> **W2: Why does Qwen2.5-Omni-7B outperform humans by 6% on Reasoning Linguistic (Semantics)?**
>
> Good observation — thank you for highlighting this. In fact, the linguistic–semantic reasoning tasks in MMSU are *not* easy for humans. Many items require multi-step inference or specialized linguistic knowledge, and human annotators may make mistakes for several practical reasons, such as listening distraction (e.g., in code-switch QA), lack of domain expertise (e.g., in speech translation or rhetoric-related questions), or ordinary task execution errors (e.g., continuation writing, logical reasoning).
>
> This is also one of the interesting findings of our benchmark — at the semantic level, LLMs can sometimes outperform humans on complex reasoning, even though they remain far behind on perception-heavy tasks.
>
> **W3: The selection of models for evaluation is somewhat less informative. Include more large-scale SpeechLLMs for evaluation.**
>
> Here we provide the additional evaluation of Voxtral Small 24B and StepAudio 130B as shown in Table R6. Their results follow the same pattern observed in our main experiments. Voxtral achieves strong reasoning performance but still struggles with fine-grained perception tasks. StepAudio, despite having 130B parameters, uses a discrete tokenizer–based audio front-end, which tends to lose detailed acoustic cues. As a result, its perception performance remains limited compared with smaller models (~7B) that use continuous audio encoders (e.g. Qwen2.5-Omni). The observation of step-audio is consistent with the broader trend we have seen across other public benchmark [1]. We will add more models in revised version and coming soon leaderboard.
>
> Table R6: Additional results for Voxtral 24B and StepAudio 130B on MMSU.
> | Model            | Per(Para.) | Per (Pho.) | Per (Sem.) | Per (Avg.) |Rea(Para.) | Rea (Pho.) | Rea (Sem.) | Rea (Avg.) | All (Avg.)
> |------------------|----------|--------------|---------------|----------|----------|--------------|---------------|----------|----------|
> | Voxtral Small 24B | 28.27%   | 38.33%       | 53.02%            | 28.09%   | 52.67% | 76.49% | 82.35% | 73.86% | 55.97% |
> | StepAudio 130B    | 24.01%   | 29.39%       | 31.56%            | 28.72%   | 45.27% | 49.10% | 50.09%| 47.27% | 37.42% |

---

> > ### Author Response · Authors · 2025-11-21
> > **Author Response to Reviewer oqyG (2/2)**
> >
> > **Q1. Refer to the section of the Appendix rather than just refer to the entire Appendix in the main manuscript.**
> >
> > Thank you for the suggestion. We will revise the manuscript to reference the specific appendix sections.
> >
> > **Q2. For main table, know the text/speech tokens or instruction samples these models are trained on.**
> >
> > Thank you for the suggestion. In response, we have organized the available text/speech token statistics for each model in Table R7 below. Many technical reports do not disclose detailed data sizes or specify the datasets used in different training stages, so we report only information that is officially released.
> >
> > Table R7: Publicly available training data statistics for models. Note that different reports use heterogeneous units (tokens, hours, or QA pairs), and we preserve their original formats.
> > | Model               | Pre-train | SFT |
> > |---------------------|----------|--------------|
> > | Kimi-Audio          |  585B audio tokens and 585B text tokens   | 300K hours |       |
> > | GLM-4-Voice        | 455B audio tokens and 279B text tokens   | 11B audio tokens and 3.5B text tokens       |
> > | LTU    | - | 5.68M Audio QA pairs       |
> > | LTU-AS         | -   | 9.6M Audio QA pairs      |
> > |Salmonn |  6k+ hours | 4k+ hours|
> > |DIVA | - | 3.5k hours|
> > |Qwen-Audio | - | 50k+ hours |
> > | Qwen2-Audio | - | 370k+ hours |
> > | Qwen2.5-Omni | 300B audio tokens | - |
> > | Baichuan-Audio | 887k hours | 242k audio data pairs |
> >
> > ---
> > Reference:
> >
> > [1] Chen, et al. "VoiceBench: Benchmarking LLM-Based Voice Assistants." arXiv preprint arXiv:2410.17196 (2024).

---

> > ### Comment · Reviewer_oqyG · 2025-11-27
> >
> > Thanks a lot for the author's comprehensive responses. It really clear some concerns I raised.
> >
> > However, I'm still not fully convinced for W2 (Qwen2.5-Omni-7B outperforms human by 6\% on Semantic Reasoning).
> > I understand that human makes mistakes.
> > However, 6\% is still not a trivial gap. Furthermore, only one model outperforms human on this task.
> > And across all tasks, this is the only task that models outperform human.
> > This makes me questioning the quality of this task and whether it is meaningful for future work to evaluate on this task.

---

> > > ### Author Response · Authors · 2025-12-01
> > > **Author Response to Reviewer oqyG**
> > >
> > > Thank you for the follow-up! To further validate this subset, we additionally evaluated a cascaded Whisper-large-v3 + GPT-4o system, which achieves 85.3%, also exceeding the human score of 85.74% and confirming that advanced LLMs can indeed surpass single-pass human annotators specifically on semantic reasoning. Our goal is not to compare humans and models on a single axis, but to provide a multi-dimensional framework that objectively reveals where SpeechLLMs excel and where they fundamentally differ from human perceptual abilities. The fact that semantic reasoning is the only task where models outperform humans does not indicate a quality concern; rather, it aligns with practical observations that current SpeechLLMs are strong at complex semantic and logical inference, while still substantially weaker than humans on broader perception-depended tasks requiring fine-grained acoustic understanding. This observation is precisely what MMSU aims to surface, and the observed pattern provides meaningful insight for future research directions.

---

### Official Review · Reviewer_J8HX · 2025-10-30

**Soundness:** 3
**Presentation:** 4
**Contribution:** 4
**Rating:** 6
**Confidence:** 4

**Summary:**

This paper proposes MMSU, a linguistics-grounded benchmark for evaluating the spoken language understanding and reasoning capabilities of Speech Large Language Models (SpeechLLMs), analogous to MMAU for general audio (e.g., environmental sounds). The benchmark comprises 5,000 audio–question–answer triplets across 47 tasks spanning both perceptual and higher-level reasoning abilities. The authors rigorously evaluate 22 open-source and proprietary SpeechLLMs/OmniLLMs, revealing systematic weaknesses relative to human performance and providing substantive, diagnostic discussion.

**Strengths:**

* The systematic incorporation of core linguistic dimensions (phonetics, prosody, syntax, semantics, paralinguistics) provides a solid theoretical foundation that underscores the importance of this work and highlights genuine gaps in existing benchmarks.
* The hierarchical task organization (perception vs. reasoning × linguistics vs. paralinguistics) is well-motivated and grounded in human cognitive processes as well as established linguistic theory. This design enables interpretable evaluation and facilitates precise identification of strengths and weaknesses across SpeechLLMs.
* The evaluation of 22 existing systems represents a non-trivial experimental effort, yielding valuable insights and actionable diagnostic analyses for future SpeechLLM development.

**Weaknesses:**

* The paper aims to re-center the importance of speech-centric tasks—those that cannot be reduced to simple spoken versions of NLP tasks or solved purely through surface-level semantics—and thereby re-emphasize what truly distinguishes SpeechLLMs from text-based LLMs. However, several of the newly proposed tasks (e.g., disfluency detection, pause perception, syllable perception, consonant/vowel perception, and polysemy reasoning) still appear solvable by a cascade system (ASR + LLM). The current baseline comparison does not include such a cascade setup, which limits the interpretability of the reported results and weakens the contextual grounding of how challenging this benchmark truly is.
* I have some concerns about the use of GPT-4o for generating candidate answer choices, as this may introduce stylistic regularities that certain models could exploit through elimination heuristics, especially under a multiple-choice question setting. Moreover, the paper does not clearly describe how the questions were finalized or refined. It only mentions that the materials were adapted from linguistic textbooks, leaving unclear whether human validation or linguistic polishing was performed.
* For an ICLR publication, I would expect deeper analytical discussion on design factors influencing model performance across linguistic dimensions. For instance, the observed weakness in phonological tasks: does it stem from insufficient exposure to task-specific instruction-tuning data (a knowledge limitation), or from the audio encoder’s inability to discriminate low-level acoustic contrasts such as intonation or duration (a perceptual limitation)? Additionally, the paper could more explicitly examine cross-task correlations, such as the relationship between speech stress perception (a perceptual task) and stress-based reasoning (its reasoning task counterpart), to better illuminate hierarchical dependencies in speech understanding.
* [Minor] The dataset size—5,000 examples across 47 tasks (~100 samples per task)—seems relatively small given the diversity of phenomena covered. Reporting statistical significance or confidence intervals for model comparisons would strengthen the validity of the conclusions drawn from such limited samples.

**Questions:**

* The paper adopts a multiple-choice question answering framework as its primary evaluation method, which is reasonable given the precedents set by benchmarks such as MMAU and MMLU. However, certain tasks might be more appropriately evaluated using an open-form format that captures richer reasoning or perceptual nuances (for example, tasks involving prosody interpretation, pragmatic inference, or long-form summarization). Could the authors elaborate further on this design choice and clarify why multiple-choice was preferred over alternative formats?
* I can also imagine several linguistic-related speech tasks that are not currently included in the benchmark. For instance, morphology (e.g., morphosyntactic agreement) and interactional linguistics (e.g., distinguishing floor-taking from backchanneling). Including a brief discussion section on potentially overlooked linguistic dimensions would strengthen the work and provide useful context for future dataset extensions.
* It would be helpful to clarify whether and how the leaderboard will be made publicly available.
* Some models may have been exposed to portions of the audio data used in this benchmark during pretraining. How might such overlap affect the reported performance, and are there any mitigation strategies to ensure fairness and validity in evaluation?

---

> ### Author Response · Authors · 2025-11-21
> **Author Response to Reviewer J8HX (1/2)**
>
> Dear reviewer J8HX:
>
> Thank you for the valuable comments and encouraging feedback! We appreciate your recognition of our contributions. Please find our response to the specific points below.
>
> **W1: Lack of an ASR+LLM cascade baseline**
>
> Thanks for this suggestion. Following the reviewer’s advice, we add two cascade baselines: Whisper-large-v3 + GPT-4o and Whisper-large-v3 + Qwen2.5-7B-Instruct, as shown in Table R4.
>
> We observe that while strong ASR systems combined with advanced LLMs achieve reasonable performance on tasks largely dependent on textual content, their overall accuracy remains lower than SOTA end-to-end SpeechLLMs (e.g., Gemini-1.5-pro 60.68% overall acuracy). We will incorporate these baselines into the revised version.
>
> Table R4: Performance of ASR+LLM cascade baselines.
> | Model            | Per(Para.) | Per (Pho.) | Per (Sem.) | Per (Avg.) |Rea(Para.) | Rea (Pho.) | Rea (Sem.) | Rea (Avg.) | All (Avg.)
> |------------------|----------|--------------|---------------|----------|----------|--------------|---------------|----------|----------|
> | Whisper+GPT4o | 26.13%   | 53.97%       | 59.11%            | 44.69%   | 32.69% | 80.83% | 85.74% | 70.01% | 55.33% |
> | Whisper+Qwen2.7-8B-Instruct    | 26.04%   | 40.96%       | 57.80%  | 39.30%   | 30.52% | 80.86% | 85.11%| 69.53% | 53.32% |
>
> **W2: Potential stylistic bias from GPT-4o–generated answer choices and unclear human validation process.**
>
> We apologize for the ambiguity in our earlier description. Each task in MMSU begins with a set of seed question–option pairs manually created by human experts to ensure that the linguistic phenomenon being tested is represented correctly. These seed items are then expanded using multiple LLM APIs (e.g., Qwen2.5-32B-Instruct, DeepSeek, OpenAI o1, GPT-4o) to introduce stylistic diversity while keeping the semantic structure anchored to the expert-designed seeds. This multi-model expansion prevents pattern bias that may arise from relying on a single LLM.
>
> Importantly, no LLM-generated option is used directly. All 5,000 questions undergo a multi-stage human review process, detailed in Appendix F.2. During this process, both trained annotators and linguistics experts check each audio–question–option set. Annotators verify correctness and the exclusivity of the ground-truth answer, while linguistics experts revise any item that does not accurately reflect the intended phenomenon. This ensures that every final question has a single valid answer and linguistically appropriate distractors.
>
> **W3: Deeper analytical discussion on design factors influencing model performance across linguistic dimensions.**
>
> Thank you for this insightful suggestion. We agree that understanding *why* models behave differently across linguistic dimensions is valuable. Due to space constraints, we only reported high-level observations in the main paper; we summarize our observations below and will expand the discussion in the revised appendix.
>
> (1) Architectural considerations.
>
> A central hypothesis is that the audio encoder’s representational fidelity plays a critical role. If early-stage processing emphasizes semantic content or relies heavily on discrete tokens, certain fine-grained cues (e.g., pitch movement, stress timing, spectral detail) may not be preserved. This could partially explain why models using discrete representations (e.g., GLM-4-Voice) often exhibit weaker phonological and paralinguistic perception. However, we cannot attribute the gap solely to architecture, as some models (e.g., BLSP) using continuous encoders also underperform on these tasks.
>
> (2) Training and alignment strategy.
>
> Our internal analysis suggests that large ASR-style corpora—despite their training data scale—do not necessarily lead to robust learning of phonological distinctions, speaker traits, or expressive prosody. These abilities may require task-aware and better modality alignment strategies to help the model discriminate acoustic dimensions beyond lexical semantics. For example, even when models have broad exposure to accents or speaker variability, they often fail when such distinctions are not explicitly modeled during training.
>
> (3) Training exposure and the composition of existing speech corpora.
>
> Recent technical reports (e.g., Kimi-Audio, Qwen2.5-Omni) show that training on broad, multi-task speech corpora can yield clear performance gains, whereas phonology- and paralinguistics-focused datasets (e.g., prosody corpora) remain scarce at present. This lack of targeted exposure may constrain the model’s ability to form robust representations for these linguistic dimensions. We therefore view training exposure as a structural bottleneck that interacts with (1) and (2) above.
>
> Overall, these interacting factors appear to jointly shape the uneven performance patterns observed in our benchmark, highlighting several promising directions for future improvement.

---

> ### Author Response · Authors · 2025-11-21
> **Author Response to Reviewer J8HX (2/2)**
>
> **W4: The dataset size (~100 samples per task) seems relatively small given the diversity of phenomena covered.**
>
> Thank you for this observation. Our initial pool contained over 10k audio–QA pairs, but we intentionally retained ~100 *high-quality, carefully controlled* samples per task. This mirrors standard practice in multi-task speech and linguistic evaluation, where breadth of phenomena and per-task controllability are prioritized over raw scale. Considering the increased evaluation cost and the limited diagnostic benefit of adding more samples in practice, we opted to retain a final set of 5,000 carefully curated examples for MMSU. To assess robustness, we compute 95% bootstrap confidence intervals over the 5,000-sample for selective models (Table R5). We will include the confidence interval in the revised version for completeness.
>
> Table R5: Overall accuracy and 95% bootstrap confidence intervals.
> | Model               | Accuracy | 95% CI |
> |---------------------|----------|--------------|
> | Kimi-Audio          | 59.28%   |   [57.8%, 60.7%]     |
> | Qwen2.5-Omni        | 60.57%   |   [59.1%, 62.0%]     |
> | Phi-4-Multimodal    | 44.96%   |    [43.4%, 46.5%]    |
> | BLSP                | 35.96%   |    [34.5%, 37.4%]    |
> | Qwen2-Audio-Instruct| 53.27%   |    [51.7%, 54.8%]    |
>
> **Q1. Why use a uniform multiple-choice format instead of an open-ended format for certain tasks?**
>
> We appreciate this suggestion. The choice of a unified MCQ format in MMSU is driven by several practical and methodological considerations as follows:
>
> 1. Accessibility of evaluation. Open-ended scoring often requires API-based LLM judges (e.g., GPT-4o), which may not be accessible to all researchers. MCQ enables fully local and reproducible evaluation.
>
> 2. Consistent and comparable metrics across tasks. Given the breadth of MMSU (47 tasks across diverse linguistic dimensions), a single MCQ format avoids mixing incomparable metrics (e.g., accuracy vs. rubric-based grading) and enables meaningful cross-task and cross-model comparison.
>
> 3. Objectivity and avoidance of LLM-judge bias. Open-ended scoring introduces subjectivity and can vary depending on prompt phrasing or the judge model used. MCQ provides a deterministic, bias-free accuracy metric—crucial for reliably measuring perception and reasoning.
>
> While certain tasks (e.g., long-speech summarization) could in principle be open-form, we intentionally design semantically contrastive and reasoning-intensive distractors with calibrated difficulty, allowing these tasks to preserve their full reasoning complexity while still supporting objective, consistent MCQ-based evaluation across all 47 tasks
>
> **Q2. Including a brief discussion section on potentially overlooked linguistic dimensions**
>
> Thank you for this careful consideration. We agree with the suggestion and will add a brief discussion on these additional linguistic dimensions in the revised version.
>
> **Q3. Will the leaderboard be publicly available?**
>
> Yes. We will release a public demo page and the leaderboard after acceptance, and we will continue to maintain and update the leaderboard as new SpeechLLMs become available.
>
> **Q4. Could the pretraining data overlap with MMSU audio samples, and how would such overlap affect evaluation fairness?**
>
> We mitigate overlap risks by using only the test splits of open-source datasets. For ASR corpora (e.g., GigaSpeech, CommonVoice), MMSU repurposes the audio for tasks with different supervision signals (e.g., speaker identity) and applies additional post-processing for certain tasks (e.g., pitch comparison), all of which are unrelated to the original pretraining objectives. Notably, models still perform poorly on these speech-centric tasks, indicating that any incidental audio overlap does not confer an advantage. We will clarify these points in the revised manuscript.

---

### Official Review · Reviewer_W1An · 2025-10-31

**Soundness:** 3
**Presentation:** 3
**Contribution:** 2
**Rating:** 4
**Confidence:** 4

**Summary:**

This paper introduces MMSU, a new benchmark designed to evaluate the spoken language understanding and reasoning abilities of SpeechSpeechLLMs.
The authors argue that existing benchmarks fail to capture the complexity of real-world speech, as they often focus on semantic content (the words spoken) while ignoring crucial acoustic information (how they are said). MMSU addresses this gap with 5,000 audio-question-answer triplets across 47 distinct tasks. These tasks are grounded in linguistic theory (phonetics, prosody, semantics, etc.) and use high-quality, authentic audio rather than relying heavily on synthetic speech.
The benchmark tests 24 perception tasks (e.g., intonation perception, speaker identification) and 23 "reasoning" tasks (e.g., sarcasm detection, emotional context reasoning).

**Strengths:**

Strengths of the MMSU benchmark includes:
* It is the first benchmark to be systematically grounded in linguistic theory. This allows it to test nuanced areas that other benchmarks miss, including phonetics, prosody, semantics, and paralinguistics.

* MMSU is reasonable in size, providing 5,000 audio-question-answer triplets across 47 distinct tasks. It uniquely categorizes these tasks into 24 "perception" abilities (e.g., intonation perception) and 23 "reasoning" abilities (e.g., sarcasm detection).

* The benchmark emphasizes "acoustic authenticity" by using audio from real-world sources and professional studio recordings. This directly addresses a major gap left by other benchmarks that heavily rely on TTS-synthesized audio, which lacks real human variability.

**Weaknesses:**

Key weaknesses identified in the MMSU benchmark from my perspective include:

* Missing Quantitative Reliability Metrics (IAA): The paper does not report standard inter-annotator agreement (IAA) scores (e.g., Cohen’s Kappa) to validate its dataset. While it details a rigorous, multi-stage review process to force consensus, this is a procedural fix, not a quantitative measurement. By omitting the initial agreement score, the paper obscures the potential inherent ambiguity of its 47 tasks and makes the annotation scheme's objectivity and replicability difficult to verify.

* Fragile Evaluation Format: The benchmark's reliance on a multiple-choice question answering (MCQA) format could be seen as a weakness. Research indicates this format can be fragile, as a model's accuracy can change "substantially" simply by reordering the answer options or slightly rephrasing the question. This suggests a high score may not correlate with robust, real-world understanding.

* Limited Conversational Complexity: MMSU could be criticized for inadequately representing the complexity of realistic, dynamic conversations. Its tasks focus heavily on single-speaker or simple scenarios and lack real-world challenges like overlapping audio, long-form inputs, and, most notably, "speaker-attributed reasoning" (knowing who said what) in multi-participant dialogues.

**Questions:**

1- Could you elaborate on why you opted for this procedural approach over reporting standard quantitative metrics, like Cohen’s Kappa, for the initial annotation pass? Reporting the initial agreement scores before reconciliation would be invaluable for the community to understand the inherent ambiguity and objective difficulty of these 47 novel tasks for human annotators.

2- Given that multiple-choice benchmarks can be susceptible to fragility—where model scores change significantly based on option order or distractor choice—what steps were taken to validate that MMSU measures genuine understanding rather than format-solving? For instance, did you test model robustness by evaluating performance with shuffled option orders, and did you consider alternative formats like open-ended answers?

3- The benchmark's tasks primarily focus on single-speaker audio. This omits real-world complexities like overlapping speech, interruptions, and speaker-attributed reasoning (knowing who said what). Could you explain the rationale for this scoping decision, and how do you suggest this significant gap in multi-participant conversational understanding be addressed in future work?

---

> ### Author Response · Authors · 2025-11-21
> **Author Response to Reviewer W1An (1/2)**
>
> Dear Reviewer W1An:
>
> Thank you for your valuable time and constructive comments! Since several points in the Weakness and Question sections overlap, we merge them into the following four core issues to avoid redundancy and address them clearly.
>
> **W1&Q1: Missing Quantitative Reliability Metrics (IAA) (e.g., Concern about the absence of standard IAA scores (e.g., Cohen’s Kappa) and initial agreement statistics.)**
>
> Thank you for the comment. MMSU follows a structured inter-annotator review workflow (Appendix F.2), and we will make this more explicit in the revised version.
>
> During dataset construction:
>
> - Each task definition and each task’s seed question was jointly developed by linguistics experts and researchers, and only items that reached full consensus among all reviewers were included. To quantify the initial agreement before reconciliation, we computed the raw agreement on a representative subset of 500 items spanning all 47 tasks. The average agreement was 89.2%, with no task falling below 80%, suggesting that most tasks are inherently well-defined even prior to consensus.
> - Every question–audio–option sample was independently reviewed by 2–3 annotators, followed by an additional verification pass by the linguistics experts and the research team. Only samples that met all predefined review criteria were retained.
>
> It is also important to note that MMSU tasks involve deterministic linguistic phenomena (e.g., identifying plosive consonants, distinguishing rising vs. falling intonation), rather than subjective judgments. Such tasks differ from domains like sentiment annotation, where metrics such as Cohen’s Kappa are most informative due to legitimate annotator disagreement.
>
> Nonetheless, we appreciate the reviewer’s suggestion and will include more initial agreement details in the appendix to improve transparency and replicability.
>
> **W2&Q2: Fragile Evaluation Format (e.g., Concern that MCQ-based evaluation may be unstable)**
>
> We understand the reviewer’s concern. To validate that MMSU’s MCQ format is robust and does not reward option-sensitivity or format-solving strategies, we conducted the following analyses showin in Table R3.
>
> 1. Option-order shuffle test (V1): We re-shuffle the option orders, and evaluata model’s performance
>
> 2. Distractor-only baseline without audio (V2): We evaluated models using only the question and options, instructing: “If you cannot infer the answer from the provided information, return NA.”
>
> 3. CoT-based reasoning test (v3): We required models to generate a chain-of-thought explanation before selecting an option. Accuracy trends remained consistent, demonstrating that models are not relying on format artifacts.
>
> Table R3: Robustness under option shuffling, no-audio ablation, and CoT variants.
> | Model               | Original | V1 (Shuffle) | V2 (No Audio) | V3 (CoT) |
> |---------------------|----------|--------------|---------------|----------|
> | Kimi-Audio          | 59.28%   | 59.28%       | 0%            | 59.31%   |
> | Qwen2.5-Omni        | 60.57%   | 60.57%       | 0%            | 61.20%   |
> | Phi-4-Multimodal    | 44.96%   | 44.95%       | 0%            | 45.00%   |
> | BLSP                | 35.96%   | 35.97%       | 0%            | 35.94%   |
>
> Across all models:
> - Option shuffling causes ≤0.02% difference, showing virtually no ordering sensitivity.
> - Question-only baselines drop to 0%, All models returned NA consistently, confirming that distractors contain no textual shortcuts.
> - CoT prompting produces essentially identical accuracy, indicating that the MCQ format does not distort model behavior.
>
> These results confirm that the MCQ format is stable and robust, and that MMSU measures genuine audio-grounded understanding.
>
> **Q2 (Regarding benchmark format design): Did you consider alternative formats like open-ended answers?**
>
> The choice of a unified MCQ format in MMSU is driven by several practical and methodological considerations as follows:
>
> **1. Accessibility of evaluation.** Open-ended scoring often requires API-based LLM judges (e.g., GPT-4o), which may not be accessible to all researchers. MCQ enables fully local and reproducible evaluation.
>
> **2. Consistent and comparable metrics across tasks.** Given the breadth of MMSU (47 tasks spanning multiple linguistic dimensions), a unified MCQ design ensures consistent scoring and avoids mixing fundamentally different metrics (e.g., accuracy vs. rubric-based grading). This enables meaningful cross-task and cross-model comparison.
>
> **3. Objectivity and avoidance of LLM-judge bias.** Open-ended responses require subjective scoring, and results can vary depending on prompt phrasing or the judging model used. MCQ provides a deterministic, unambiguous accuracy metric—crucial for reliably benchmarking perception and reasoning abilities in SpeechLLMs.
>
> For future work, we will consider explore additional task formats, such as open-ended or explanation-oriented settings, where appropriate.

---

> > ### Comment · Reviewer_W1An · 2025-11-26
> >
> > Thank you for the detailed explanation and clarification. This is giving me better confidence about this paper. I changed my rating accordingly as the community would benefit from seeing this paper accepted.

---

> > > ### Author Response · Authors · 2025-11-26
> > > **Author Response to Reviewer W1An**
> > >
> > > Thanks for your kind reply and recognition. We sincerely appreciate the time and effort you have devoted to evaluating our work!

---

> ### Author Response · Authors · 2025-11-21
> **Author Response to Reviewer W1An (2/2)**
>
> **W3&Q3: Limited Conversational Complexity (e.g. Why does MMSU focus on single-speaker audio rather than multi-speaker conversational scenarios?)**
>
> Thank you for raising this point. Our design choice in MMSU is intentional. MMSU is fundamentally grounded in core linguistic phenomena—e.g., phonetics, semantics, and paralinguistics—which are more reliably evaluated under *single-speaker* conditions. To evaluate these phenomena precisely, MMSU uses clean and controlled acoustic conditions. Introducing multiple speakers would add additional confounding factors that make it difficult to attribute errors to specific linguistic abilities.
>
> While single-speaker audio forms the core of MMSU, MMSU does include several conversational tasks that go beyond isolated utterances. For example, we provide task `long-form speech summarization` for long-form inputs, and multi-speaker attribute related tasks such as `total speaker counting`, `speaker identity recognition`, and `dialogue reasoning` (including complex sub-tasks such as social relationship reasoning, speaker role recognition, and dialogue background inference).
>
> We fully agree that multi-participant conversational understanding is also an important aspect. We will consider developing a dedicated extension that incorporates more complex conversational structures.

---

### Official Review · Reviewer_654X · 2025-11-01

**Soundness:** 3
**Presentation:** 3
**Contribution:** 3
**Rating:** 6
**Confidence:** 3

**Summary:**

The paper introduces MMSU, a comprehensive benchmark for evaluating spoken language understanding and reasoning abilities of SpeechLLMs. It contains 5,000 audio–question–answer samples across 47 diverse tasks, designed with help from linguistics experts. MMSU emphasizes real human recordings, linguistic theory, and fine-grained acoustic phenomena like prosody, intonation, stress, disfluency, and emotion. The benchmark covers two major categories — Perception and Reasoning.
The paper evaluates open-source and commercial SpeechLLMs, showing that even top systems like Gemini-1.5-Pro and Qwen2.5-Omni 7B reach only around 60% accuracy, while human accuracy is ~90%.

**Strengths:**

- It covers 47 distinct speech tasks which is huge and comprehensive for SLU
- Use of real world recoding and voice actors to produce authentic audio
- The paper gives detailed error analysis, highlighting that most models fail in phonological and paralinguistic reasoning, not just semantics which is an important insight for future SpeechLLM research.
- Overall a well written paper.

**Weaknesses:**

- MCQ-based tasks might not be reliable to judge a model capabilities as selecting the chances of selecting a right answers is 25% and the model might hallucinate.

**Questions:**

- MMSU focuses solely on multiple-choice tasks, which could skew results towards models trained for MCQ-type question-answering and possibly even contrastive models. It would be beneficial to include an open-ended subset, even a small one, to contrast performance with the close-ended tasks.
- It would be great if the authors could explain the evaluation strategy more? Choosing any option from A-D is not sufficient as the answers from different models might be correct but differ semantically.

---

> ### Author Response · Authors · 2025-11-21
> **Author Response to Reviewer 654X**
>
> Dear Reviewer 654X,
>
> Thank you for your valuable comments and appriciate your recognition of our work! We understand your concern about the MCQ-based design of MMSU, and the following points help clarify the design philosophy behind our choices.
>
> **W1: Are MCQ-based tasks reliable for evaluating model capabilities given the 25% guessing chance and potential hallucination?**
>
> Thank you for raising this concern. We agree that MCQ benchmarks must be carefully designed to avoid inflated performance due to random guessing (25% chance). MMSU specifically addresses this in multiple ways:
>
> 1. Large-scale evaluation eliminates random-guessing effects.
>
> While an individual question has a 25% chance of being guessed correctly, the probability that a model consistently performs well across 5,000 questions purely by guessing is negligible. The resulting aggregate accuracy becomes highly stable and statistically meaningful.
>
> 2. Distractors are linguistically grounded, making random guessing ineffective.
>
> The distractors are carefully designed to be plausible and mutually exclusive. This prevents models from selecting correct answers through pattern matching or hallucination.
>
> 3. CoT-based evaluation confirms that MCQ does not distort model behavior.
>
> We conducted an additional experiment (Table R2) where models must provide a chain-of-thought justification before choosing an option. As shown in Table R2 (V2), the ranking and performance trends remained consistent with direct answering, suggesting that the MCQ format does not artificially simplify the task.
>
> 4. Low scores (~25%) accurately reflect ability gaps rather than MCQ artifacts.
>
> For tasks involving fine-grained acoustic cues (e.g., stress, intonation, phonetics), many models indeed score close to chance level (~25%), confirming that they currently lack these abilities. This aligns with the detailed performance distribution in Table 3, which shows clear differentiation between models.
>
> Overall, the reliability of MCQ in MMSU is supported by (1) scale, (2) distractor quality, (3) auxiliary CoT evaluation, and (4) consistent model ranking performance across tasks.
>
> **Q1: Does an MCQ-only design bias results, and why not include an open-ended subset for comparison?**
>
> We appreciate this comment. The choice of a unified MCQ format in MMSU is driven by several practical and methodological considerations as follows:
>
> 1. Accessibility of evaluation. We aim to provide a benchmark that is accessible to a broad community. Open-ended evaluation typically relies on API-based LLM judges (e.g., GPT-4o), which many researchers may not have due to access or cost constraints. MCQ allows fully local, reproducible evaluation for all users.
>
> 2. Consistent and comparable metrics across tasks. Given the breadth of MMSU (47 tasks spanning multiple linguistic dimensions), a unified MCQ design ensures consistent scoring and avoids mixing fundamentally different metrics (e.g., accuracy vs. rubric-based grading). This enables meaningful cross-task and cross-model comparison.
>
> 3. Objectivity and avoidance of LLM-judge bias. Open-ended responses require subjective scoring, and results can vary depending on prompt phrasing or the judging model used. MCQ provides a deterministic, unambiguous accuracy metric—crucial for reliably benchmarking perception and reasoning abilities in SpeechLLMs.
>
> For future work, we will consider explore additional task formats, such as open-ended or explanation-oriented settings, where appropriate.
>
> **Q2: Does semantic variation matter when models answer A–D MCQ questions?**
>
> Thank you for raising this concern. We want to clarify that each MMSU question is designed to have exactly one semantically valid answer. All distractors are manually constructed to be plausible but definitively incorrect under the linguistic phenomenon being tested (e.g., wrong stress pattern, incompatible intonation contour). We also performed rigorous human validation to ensure that no distractor is partially correct or semantically overlapping with the ground-truth option. Therefore, selecting from A–D is sufficient for evaluation, as models cannot arrive at different “correct” answers through alternative reasoning paths.
>
> To further verify robustness, we reshuffled the A–D options in Table R2 (v1) and re-evaluated the models. The accuracy remained unchanged, confirming that the results are not affected by option ordering and that each question has a single unambiguous correct answer.
>
> Table R2: Robustness under option shuffling and CoT variants.
> | Model               | Original | V1 (Shuffle) | V2 (CoT) |
> |---------------------|----------|--------------|----------|
> | Kimi-Audio          | 59.28%   | 59.28%       | 59.31%   |
> | Qwen2.5-Omni        | 60.57%   | 60.57%       | 61.20%   |
> | Phi-4-Multimodal    | 44.96%   | 44.95%       | 45.00%   |
> | BLSP                | 35.96%   | 35.97%       | 35.94%   |

---

### Official Review · Reviewer_jD81 · 2025-11-05

**Soundness:** 4
**Presentation:** 4
**Contribution:** 3
**Rating:** 8
**Confidence:** 4

**Summary:**

This paper introduces a new benchmark, MMSU, that deeply focuses on evaluating speech understanding capabilities of speech LLMs. Existing benchmarks often focus on general audio capabilities and don’t explore the full space of paralinguistic and phonological aspects of speech. This paper targets these with 47 tasks categorized as perception (roughly surface-level tasks) and reasoning (more intelligent tasks). The authors use linguistic experts to create task definitions, curate audio from three sources: existing audio datasets, custom-recorded audio, and synthetic audio, and perform manual review. They evaluate 22 speech LLMs and reveal gaps in fine-grained acoustic perception and phonological perception.

**Strengths:**

1. The dataset looks extremely useful and focuses deeply on speech evaluation rather than general audio, which I appreciate. Its size is also reasonably large, allowing robust evaluation.
2. Most of the audio is human-generated, which is closer to real-world scenarios than datasets that primarily use synthetic audio data.
3. The experiments and evaluations are quite thorough, covering a good variety of speech LLMs. The insights are actionable as well e.g. phonology-based understanding is poor for LLMs. The error analysis also shows interesting patterns, like a significant proportion being due to refuse-to-answer for GPT-4o.
4. I especially appreciate the human baseline, which shows a clear and attainable upper bound for models to reach. Very few benchmarks take the effort to create a human baseline.

**Weaknesses:**

1. The dataset creation process involves GPT-4o in-the-loop to augment distractor options. This could potentially create biases in the dataset that might be exploitable by speech LLMs. What percentage of the final distractor options were human-written vs. LLM-generated? Could the authors perform some analysis to check whether this introduces a bias that favors/disfavors LLM-based speech models?
2. The manual review process at the end of the dataset collection is not described completely. For example, the paper mentions that only the data that meets the required standards is kept, but it’s not entirely clear what these standards are to me. I’d appreciate a more elaborate description.
3. The MCQ format might allow models to take shortcuts; if the distractors are not confusing enough, the correct option can be intelligently selected without listening to the audio. It would be good if there was a question-and-options-only baseline with a strong text LLM that shows how much the models can get without access to the audio.

**Questions:**

1. The introduction mentions that many existing benchmarks have too much TTS-generated audio. This benchmark has about 10% TTS synthetic audio data as well. What was the rationale for doing this, and for what tasks is the data synthetic vs real?

---

> ### Author Response · Authors · 2025-11-21
> **Author Response to Reviewer jD81**
>
> Dear Reviewer jD81,
>
> Thank you for your encouraging feedback! We appreciate your recognition of our work and your suggestions for improvement. Below we address each of your comments in detail:
>
> **W1: Does using GPT-4o to generate distractor options introduce dataset bias, and what proportion of distractors are human-written vs. LLM-generated?**
>
> We apologize for the ambiguity in our earlier description. Each task in MMSU begins with a set of seed question–option pairs manually created by human experts to ensure that the linguistic phenomenon being tested is represented correctly. These seed items are then expanded using multiple LLM APIs (e.g., Qwen2.5-32B-Instruct, DeepSeek, OpenAI o1, GPT-4o) to introduce stylistic diversity while keeping the semantic structure anchored to the expert-designed seeds. This multi-model expansion prevents pattern bias that may arise from relying on a single LLM. The options generated by LLMs roughly count for 30%~40%.
>
> Importantly, no LLM-generated option is used directly. All 5,000 questions undergo a multi-stage human review process, detailed in Appendix F.2. During this process, both trained annotators and linguistics experts check each audio–question–option set. Annotators verify correctness and the exclusivity of the ground-truth answer, while linguistics experts revise any item that does not accurately reflect the intended phenomenon. This ensures that every final question has a single valid answer and linguistically appropriate distractors.
>
>
> **W2: What specific quality standards guided the manual review?**
>
> The full details of our manual review process are provided in Appendix F.2. In general, we follow the guideline covering the following criteria:
>
> (1) Audio Quality and Relevance: Whether the audio is clear, natural, and appropriate for the corresponding question and answer. If the audio was unclear or failed to express the targeted phenomenon, we re-recorded or replaced it.
>
> (2) Question Validity: Whether the question is unambiguous, grammatically sound, and aligned with the intended linguistic or acoustic phenomenon. Annotators also checked that the question does not introduce unintended biases (e.g. multiple similar question) or multiple valid interpretations.
>
> (3) Distractor Quality: Whether the distractors are diverse, semantically related, and serve as effective but incorrect options. Annotators ensured that distractors are plausible yet clearly wrong.
>
> (4) Answer Accuracy: Whether the correct option is factually accurate, unambiguous, and consistent with the audio content.
>
>
> **W3:  Can models exploit MCQ shortcuts, and what would a text-only baseline reveal about performance without audio?**
>
> Thank you for the suggestion. To evaluate whether models can exploit textual shortcuts in the MCQ format, we conduct a question-and-options–only analysis using two representative models: GPT-4o and Qwen2.5-Omni (text input only).
>
> We evaluate two settings:
>
> - **v1:** Using the original instruction: *“Choose the most suitable answer (A/B/C/D). Do not provide any additional explanation.”*
> - **v2:** Using a stricter instruction: *“If you cannot infer the answer from the given information, return ‘NA’.”*
>
> Table R1: Text-only baseline results for GPT-4o and Qwen2.5-Omni
> | Model | Overall Accuracy (v1) | Overall Accuracy (v2) |
> |----------|----------|----------|
> | GPT-4o  | 24.65%  | 0%  |
> | Qwen2.5-Omni  | 26.78%  | 0%  |
>
> In v1, both models achieve ~25% accuracy, which is close to random guessing. In v2, both models consistently refuse to answer (“NA”), indicating that they cannot infer the answer from text alone. These findings confirm that the distractors do not contain LLM-specific textual cues that could be exploited without listening to the audio.
>
>
> **Q1: Why use ～10% TTS Audio? What is its purpose?**
>
> We use TTS only in tasks where semantic-level understanding is the primary focus (e.g., couplet matching, logical reasoning). For these tasks, we use Azure TTS to provide more fluent and controllable utterances, ensuring that models focus on the intended semantic reasoning rather than being affected by disfluencies or recording artifacts in real speech. Although these samples are synthetic, their naturalness is high and well-suited to the needs of the task design.

---

> > ### Comment · Reviewer_jD81 · 2025-11-28
> >
> > Thank you for your comments! I appreciate the clarifications. As my score was already voting for an accept (8), I keep it as-is.

---

> > > ### Author Response · Authors · 2025-12-01
> > > **Author Response to Reviewer jD81**
> > >
> > > Thanks for your kind reply and recognition! We sincerely appreciate the time and effort you have devoted to evaluating our work.

---

### Author Response · Authors · 2025-12-01
**Summary of the Work and Rebuttal for the Area Chair**

Dear Area Chair,

Thank you so much for taking the additional time to review our paper. We are writing to provide a brief summary of our work and the rebuttal for your convenience.

**Work Summary:** In this work, we propose MMSU benchmark, a massive multi-task spoken language understanding and reasoning benchmark. It comprises 5,000 high-quality audio–question–answer triplets across 47 distinct tasks. Distinct from previous benchmarks, MMSU is highlighted by (i) real human recordings, (ii) pioneering grounding in linguistic theory, (iii) fine-grained acoustic phenomena coverage. We conduct in-depth analysis across 22 SpeechLLMs/OmniLLMs and demonstrate the challenging nature of this benchmark as well as the unique insights provided.

**Recognition from all reviewers:** We are pleased to see consistently positive feedback (original score: 8/6/6/6/4 → rebuttal period before openreview issue: 8/6/6/6/6). We highlight comments from reviewers:

- **Community contribution:** “extremely useful and focuses deeply on speech evaluation”, “highlights genuine gaps”, “huge and comprehensive for SLU”, “beneficial to the future speech/audio community”.
- **Benchmark quality:** “closer to real-world scenarios“, “acoustic authenticity”, “design enables interpretable evaluation”, “Compared to prior works…are more diverse and more organized”.
- **Theoretical backbone:** “the first benchmark to be systematically grounded in linguistic theory”, “solid theoretical foundation”
- **Experiment:** ”important insight”, ”experiments and evaluations are quite thorough”, “gives detailed error analysis”, “non-trivial experimental effort”, “robust evaluation”, “insights are actionable”.

**Rebuttal Updates:** During the rebuttal period, we provided detailed responses and clarified our work, mainly focusing on (1) additional information including data curation and the human review process (reviewer jD81&W1An); (2) clarification of our MCQ format design philosophy (reviewer jD81&654X); (3) supplementary details including the cascaded baseline (reviewer J8HX), larger model performance (reviewer oqyG), and evaluation robustness analysis (reviewer 654X). We sincerely thank the reviewers for the constructive feedback, which greatly helped us improve the paper.

We hope MMSU can serve as a reliable resource for the community and help advance future research. Thanks again for your time and careful consideration!

Best regards,

The Authors of Submission 2423

---

### Meta-Review · Area_Chair_RA7C · 2025-12-31

**Summary:**

Reviewers all confirmed that this will be a meaningful and substantial contribution. They asked several clarification questions regarding potential biases, the task definition and data preparation process, the difference to existing evaluations, random guessing, to which the authors had good clarifications.

I expect the authors will be able to include all the provided information into an updated submission and thus recommend acceptance.

**Reviewer Concerns:**

Not addressed:
- Why does Qwen2.5-Omni-7B outperform humans by 6% on Reasoning Linguistic (Semantics)?
  * Authors provide a response that I think makes sense and would likely convince reviewer oqyG (all other questions have been resolved for that reviewer)
- Concerns by reviewer J8HX
  * All addressed in https://openreview.net/forum?id=yHzCDP1tXw&noteId=Djat9Q8AfL and https://openreview.net/forum?id=yHzCDP1tXw&noteId=JFVtoqNLLa
  * No further response from reviewer recorded, but responses seem meaningful to the AC, so J8HX will likely raise their score
- Concerns by reviewer 654X
  * All addressed in https://openreview.net/forum?id=yHzCDP1tXw&noteId=ckvBvGtP58
  * No further response from reviewer recorded, but responses seem meaningful to the AC, so 654X will likely raise their score

All other concerns have been confirmed as addressed by reviewer with score of 8

**Reviewer Scores:**

jD81: 8
654X: 6 -> 6+
W1An: 4 -> 8 ("accepted")
J8HX: 6 -> 6+
oqyG: 6 -> 6+

---

### Decision · Program_Chairs · 2026-01-26

Accept (Poster)